# Planning from Pixels in Environments with Combinatorially Hard Search Spaces

**Marco Bagatella**
Max Planck Institute for Intelligent Systems
Tübingen, Germany
mbagatella@tue.mpg.de

**Mirek Olšák**
Computer Science Department
University Innsbruck, Austria
mirek@olsak.net

**Michal Rolínek**
Max Planck Institute for Intelligent Systems
Tübingen, Germany
michal.rolinek@tue.mpg.de

**Georg Martius**
Max Planck Institute for Intelligent Systems
Tübingen, Germany
georg.martius@tue.mpg.de

## Abstract

The ability to form complex plans based on raw visual input is a litmus test for current capabilities of artificial intelligence, as it requires a seamless combination of visual processing and abstract algorithmic execution, two traditionally separate areas of computer science. A recent surge of interest in this field brought advances that yield good performance in tasks ranging from arcade games to continuous control; these methods however do not come without significant issues, such as limited generalization capabilities and difficulties when dealing with combinatorially hard planning instances. Our contribution is two-fold: (i) we present a method that learns to represent its environment as a latent graph and leverages state reidentification to reduce the complexity of finding a good policy from exponential to linear (ii) we introduce a set of lightweight environments with an underlying discrete combinatorial structure in which planning is challenging even for humans. Moreover, we show that our methods achieves strong empirical generalization to variations in the environment, even across highly disadvantaged regimes, such as "one-shot" planning, or in an offline RL paradigm which only provides low-quality trajectories.

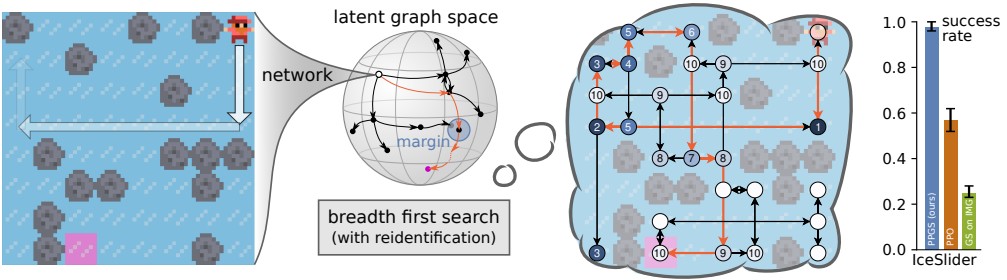

Figure 1: Planning from Pixels with Graph Search. Our method leverages learned latent dynamics to efficiently build and search a graph representation of the environment. Resulting policies show unrivaled performance across a distribution of hard combinatorial tasks.

35th Conference on Neural Information Processing Systems (NeurIPS 2021).

# 1   Introduction

Decision problems with an underlying combinatorial structure pose a significant challenge for a learning agent, as they require both the ability to infer the true low-dimensional state of the environment and the application of abstract reasoning to master it. A traditional approach for common logic games, given that a simulator or a model of the game are available, consists in applying a graph search algorithm to the state diagram, effectively simulating several trajectories to find the optimal one. As long as the state space of the game grows at a polynomial rate with respect to the planning horizon, the solver is able to efficiently find the optimal solution to the problem. Of course, when this is not the case, heuristics can be introduced at the expense of optimality of solutions.

Learned world models [17, 18] can learn to map complex observations to a lower-dimensional latent space and retrieve an approximate simulator of an environment. However, while the continuous structure of the latent space is suitable for training reinforcement learning agents [12, 19] or applying heuristic search algorithms [38], it also prevents a straightforward application of simpler graph search techniques that rely on identifying and marking visited states.

Our work follows naturally from the following insight: a simple graph search might be sufficient for solving visually complex environments, as long as a world model is trained to realize a suitable structure in the latent space. Moreover, the complexity of the search can be reduced from exponential to linear by reidentifying visited latent states.

The method we propose is located at the intersection between classical planning, representation learning and model-based reinforcement learning. It relies on a novel low-dimensional world model trained through a combination of opposing losses without reconstructing observations. We show how learned latent representations allow a dynamics model to be trained to high accuracy, and how the dynamics model can then be used to reconstruct a *latent graph* representing environment states as vertices and transitions as edges. The resulting latent space structure enables powerful graph search algorithms to be deployed for planning with minimal modifications, solving challenging combinatorial environments from pixels. We name our method **PPGS** as it **P**lans from **P**ixels through **G**raph **S**earch.

We design PPGS to be capable of generalizing to unseen variations of the environment, or equivalently across a distribution of *levels* [13]. This is in contrast with traditional benchmarks [7], which require the agent to be trained and tested on the same fixed environment.

We can describe the main contributions of this paper as follows: first, we introduce a suite of environments that highlights a weakness of modern reinforcement learning approaches, second, we introduce a simple but principled world model architecture that can accurately learn the latent dynamics of a complex system from high dimensional observations; third, we show how a planning module can simultaneously estimate the latent graph for previously unseen environments and deploy a breadth first search in the latent space to retrieve a competitive policy; fourth, we show how combining our insights leads to unrivaled performance and generalization on a challenging class of environments.

# 2   Method

For the purpose of this paper, each environment can be modeled as a family of fully-observable deterministic goal-conditioned Markov Decision Processes with discrete actions, that is the 6-tuples $\{(S, A, T, G, R, \gamma)_i\}_{1...n}$ where $S_i$ is the state set, $A_i$ is the action set, $T_i$ is a transition function $T_i : S_i \times A_i \to S_i$, $G_i$ is the goal set and $R_i$ is a reward function $R_i : S_i \times G_i \to \mathbb{R}$ and $\gamma_i$ is the discount factor. We remark that each environment can also be modeled as a BlockMDP [14] in which the context space $\mathcal{X}$ corresponds to the state set $S_i$ we introduced.

In particular, we deal with families of procedurally generated environments. We refer to each of the $n$ elements of a family as a *level* and omit the index $i$ when dealing with a generic level. We assume that state spaces and action spaces share the same dimensionality across all levels, that is $|S_i| = |S_j|$ and $|A_i| = |A_j|$ for all $0 \leq i, j \leq n$.

In our work the reward simplifies to an indicator function for goal achievement $R(s, g) = \mathbf{1}_{s=g}$ with $G \subseteq S$. Given a goal distribution $p(g)$, the objective is that of finding a goal-conditioned policy $\pi_g$ that maximizes the return

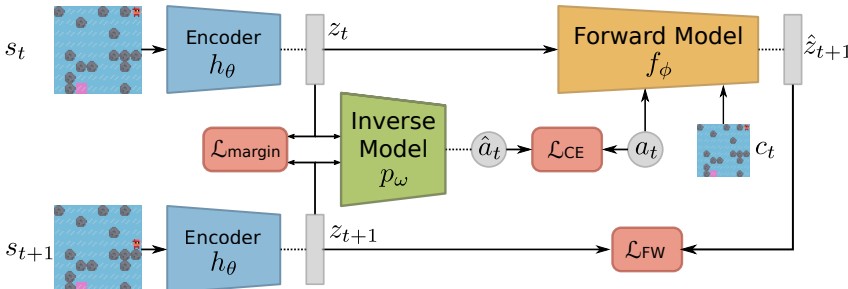

Figure 2: Architecture of the world model. A convolutional encoder extracts latent state representations from observations, while a forward model and an inverse model reconstruct latent dynamics by predicting state transitions and actions that cause them. The notation is introduced in Sec. 2.1

$$\mathcal{J}_\pi = \mathop{\mathbb{E}}_{g \sim p(g)} \left[ \mathop{\mathbb{E}}_{\tau \sim p(\tau | \pi_g)} \sum_t \gamma^t R(s_t, g) \right] \tag{1}$$

where $\tau \sim p(\tau | \pi_g)$ is a trajectory $(s_t, a_t)_{t=1}^T$ sampled from the policy.

Our environments of interest should challenge both perceptive and reasoning capabilities of an agent. In principle, they should be solvable through extensive search in hard combinatorial spaces. In order to master them, an agent should therefore be able to (i) identify pairs of bisimilar states [43], (ii) keep track of and reidentify states it has visited in the past and (iii) produce highly accurate predictions for non-trivial time horizons. These factors contribute to making such environments very challenging for existing methods. Our method is designed in light of these necessities; it has two integral parts, the world model and the planner, which we now introduce.

## 2.1 World Model

The world model relies solely on three jointly trained function approximators: an encoder, a forward model and an inverse model. Their overall orchestration is depicted in Fig. 2 and described in the following.

### 2.1.1 Encoder

Mapping highly redundant observations from an environment to a low-dimensional state space $Z$ has several benefits [17, 18]. Ideally, the projection should extract the compressed "true state" of the environment and ignore irrelevant visual cues, discarding all information that is useless for planning. For this purpose, our method relies on an *encoder* $h_\theta$, that is a neural function approximator mapping each observed state $s \in S$ and a low-dimensional representation $z \in Z$ (*embedding*). While there are many suitable choices for the structure of the latent space $Z$, we choose to map observations to points on an $d$-dimensional hypersphere taking inspiration from Liu et al. [29].

### 2.1.2 Forward Model

In order to plan ahead in the environment, it is crucial for an agent to estimate the transition function $T$. In fact, if a mapping to a low-dimensional latent space $Z$ is available, learning directly the projected transition function $T_Z : Z \times A \to Z$ can be largely beneficial [17, 18]. The deterministic latent transition function $T_Z$ can be learned by a neural function approximator $f_\phi$ so that if $T(s_t, a_t) = s_{t+1}$, then $f_\phi(h_\theta(s_t), a_t) := f_\phi(z_t, a_t) = h_\theta(s_{t+1})$. We refer to this component as *forward model*. Intuitively, it can be trained to retrieve the representation of the state of the MDP at time $t + 1$ given the representation of the state and the action taken at the previous time step $t$.

Due to the Markov property of the environment, an initial state embedding $z_t$ and the action sequence $(a_t, \dots, a_{t+k})$ are sufficient to to predict the latent state at time $t + k$, as long as $z_t$ successfully captures all relevant information from the observed state $s_t$. The amount of information to be

embedded in $z_t$ and to be retained in autoregressive predictions is, however, in most cases, prohibitive. Take for example the case of a simple maze: $z_t$ would have to encode not only the position of the agent, but, as the predictive horizon increases, most of the structure of the maze.

**Invariant Structure Recovery** To allow the encoder to only focus on local information, we adopt an hybrid forward model which can recover the invariant structures in the environment from previous observations. The function that the forward model seeks to approximate can then include an additional input: $f_\phi(z_t, a_t, s_c) = z_{t+1}$, where $s_c \in S$ is a generic observation from the same environment and level. Through this context input the forward model can retrieve information that is constant across time steps (e.g. the location of walls in static mazes). In practice, we can use randomly sampled observation from the same level during training and use the latest observation during evaluation. This choice allows for more accurate and structure-aware predictions, as we show in the ablations in Suppl. A.

Given a trajectory $(s_t, a_t)_{t=1}^T$, the forward model can be trained to minimize some distance measure between state embeddings $(z_{t+1})_{1...T-1} = (h_\theta(s_{t+1}))_{1...T-1}$ and one-step predictions $(f_\phi(h_\theta(s_t), a_t, s_c))_{1...T-1}$. In practice, we choose to minimize a Monte Carlo estimate of the expected Euclidean distance over a finite time horizon, a set of trajectories and a set of levels. When training on a distribution of levels $p(l)$, we extract $K$ trajectories of length $H$ from each level with a uniform random policy $\pi$ and we minimize

$$\mathcal{L}_{\text{FW}} = \mathop{\mathbb{E}}_{l \sim p(l)} \left[ \frac{1}{H-1} \sum_{h=1}^{H-1} \mathop{\mathbb{E}}_{a_h \sim \pi} \left[ \|f_\phi(z_h^l, a_h, s_c^l) - z_{h+1}^l\|_2^2 \right] \right] \tag{2}$$

where the superscript indicates the level from which the embeddings are extracted.

### 2.1.3 Inverse Model and Collapse Prevention

Unfortunately, the loss landscape of Equation 2 presents a trivial minimum in case the encoder collapses all embeddings to a single point in the latent space. As embeddings of any pair of states could not be distinguished in this case, this is not a desirable solution. We remark that this is a known problem in metric learning and image retrieval [8], for which solutions ranging from siamese networks [9] to using a triplet loss [22] have been proposed.

The context of latent world models offers a natural solution that isn't available in the general embedding problem, which consists in additionally training a probabilistic *inverse model* $p_\omega(a_t \mid z_t, z_{t+1})$ such that if $T_Z(z_t, a_t) = z_{t+1}$, then $p_\omega(a_t \mid z_t, z_{t+1}) > p_\omega(a_k \mid z_t, z_{t+1}) \forall a_k \neq a_t \in A$. The inverse model, parameterized by $\omega$, can be trained to predict the action $a_t$ that causes the latent transition between two embeddings $z_t, z_{t+1}$ by minimizing multi-class cross entropy.

$$\mathcal{L}_{\text{CE}} = \mathop{\mathbb{E}}_{l \sim p(l)} \left[ \frac{1}{H-1} \sum_{h=1}^{H-1} \mathop{\mathbb{E}}_{a_h \sim \pi} \left[ -\log p_\omega(a_h \mid z_h^l, z_{h+1}^l) \right] \right]. \tag{3}$$

Intuitively, $\mathcal{L}_{\text{CE}}$ increases as embeddings collapse, since it becomes harder for the inverse model to recover the actions responsible for latent transitions. For this reason, it mitigates unwanted local minima. Moreover, it is empirically observed to enforce a regular structure in the latent space that eases the training procedure, as argued in Sec. A of the Appendix. We note that this loss plays a similar role to the reconstruction loss in Hafner et al. [18]. However, $\mathcal{L}_{CE}$ does not force the encoder network to embed information that helps with reconstructing irrelevant parts of the observation, unlike training methods relying on image reconstruction [11, 17–20].

While $\mathcal{L}_{\text{CE}}$ is sufficient for preventing collapse of the latent space, a discrete structure needs to be recovered in order to deploy graph search in the latent space. In particular, it is still necessary to define a criterion to reidentify nodes during the search procedure, or to establish whether two embeddings (directly encoded from observations or imagined) represent the same true low-dimensional state.

A straightforward way to enforce this is by introducing a margin $\varepsilon$, representing a desirable minimum distance between embeddings of non-bisimilar states [43]. A third and final loss term can then be introduced to encourage margins in the latent space:

$$\mathcal{L}_{\text{margin}} = \mathop{\mathbb{E}}_{l \sim p(l)} \left[ \frac{1}{H-1} \sum_{h=1}^{H-1} \max\left(0, 1 - \frac{\|z_{h+1}^l - z_h^l\|_2^2}{\varepsilon^2}\right) \right]. \tag{4}$$

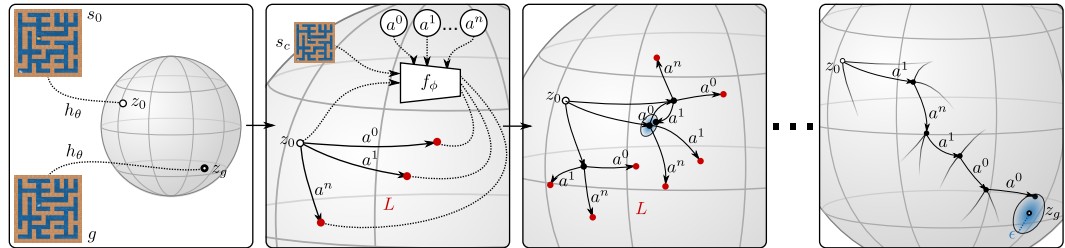

Figure 3: Overview of latent-space planning. One-shot planning is possible by (i) embedding the current observation and goal to the latent space and (ii) iteratively growing a latent graph until a vertex is reidentified with the goal.

We then propose to reidentify two embeddings as representing the same true state if their Euclidean distance is less than $\frac{\varepsilon}{2}$.

Adopting a latent margin effectively constrains the number of margin-separated states that can be represented on an hyperspherical latent space. However, this quantity is lower-bounded by the kissing number [41], that is the number of non-overlapping unit-spheres that can be tightly packed around one $d$ dimensional sphere. The kissing number grows exponentially with the dimensionality $d$. Thus, the capacity of our $d$-dimensional unit sphere latent space ($d = 16$ in our case with margin $\varepsilon = 0.1$) is not overly restricted.

The world model can be trained jointly and end-to-end by simply minimizing a combination of the three loss functions:

$$\mathcal{L} = \alpha\mathcal{L}_{\text{FW}} + \beta\mathcal{L}_{\text{CE}} + \mathcal{L}_{\text{margin}}. \tag{5}$$

To summarize, the three components are respectively encouraging accurate dynamics predictions, regularizing latent representations and enforcing a discrete structure for state reidentification.

## 2.2 Planning Regimes

A deterministic environment can be represented as a directed graph $G$ whose vertices $V$ represent states $s \in S$ and whose edges $E$ encode state transitions. An edge from a vertex representing a state $s \in S$ to a vertex representing a state $s' \in S$ is present if and only if $T(s, a) = s'$ for some action $a \in A$, where $T$ is the state transition function of the environment. This edge can then be labelled by action $a$. Our planning module relies on reconstructing the *latent graph*, which is a projection of graph $G$ to the latent state $Z$.

In this section we describe how a latent graph can be build from the predictions of the world model and efficiently searched to recover a plan, as illustrated in Fig. 3. This method can be used as a one-shot planner, which only needs access to a visual goal and the initial observation from a level. When iterated and augmented with online error correction, this procedure results in a powerful approach, which we refer to as *full planner*, or simply as PPGS.

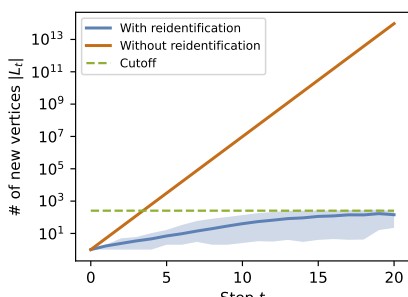

Figure 4: Number of leaf vertices when planning in ProcgenMaze, averaged over 100 levels, with 90% confidence intervals.

**One-shot Planner** Breadth First Search (BFS) is a graph search algorithm that relies on a LIFO queue and on marking visited states to find an optimal path $O(V + E)$ steps. Its simplicity makes it an ideal candidate for solving combinatorial games by exploring their latent graph. If the number of reachable states in the environment grows polynomially, the size of the graph to search will increase at a modest rate and the method can be applied efficiently.

We propose to execute a BFS-like algorithm on the latent graph, which is recovered by autoregressively simulating all transitions from visited states. As depicted in Fig. 3, at each step, the new set of leaves $L$ is retrieved by feeding the leaves from the previous iteration through the forward model $f_\phi$. The efficiency of the search process can be improved as shown in Fig. 4, by exploiting the margin $\varepsilon$ enforced by equation 4 to reidentify states and identify loops in the latent graph. We now provide a simplified description of the planning method in Algorithm 1, while details can be found in Suppl. C.2.

---

**Algorithm 1** Simplified one-shot PPGS

---

**Input:** Initial observed state $s_1$, visual goal $g$, model parameters $\theta, \phi$

1: $z_1, z_g = h_\theta(s_1), h_\theta(g)$        $\triangleright$ project to latent space $Z$
2: $L, V = \{z_1\}$        $\triangleright$ sets of leaves and visited vertices
3: **for** $T_{MAX}$ steps **do**
4:      $L = \{f_\phi(z, a, s_1) : \exists z \in L, a \in A\}$        $\triangleright$ grow graph
5:      **if** $z^* \in L$ can be reidentified with $z_g$ **then**
6:          **return** action sequence from $z_1$ to $z^*$
7:      **end if**
8:      $L = L \setminus V$        $\triangleright$ reidentify and discard visited vertices (details in Suppl. C.2)
9:      $V = V \cup L$        $\triangleright$ update visited vertices
10: **end for**

---

**Full Planner** The one-shot variant of PPGS largely relies on highly accurate autoregressive predictions, which a learned model cannot usually guarantee. We mitigate this issue by adopting a model predictive control-like approach [15]. PPGS recovers an initial guess on the best policy $(a_i)_{1,\ldots,n}$ simply by applying one-shot PPGS as described in the previous paragraph and in Algorithm 2. It then applies the policy step by step and projects new observations to the latent space. When new observations do not match with the latent trajectory, the policy is recomputed by applying one-shot PPGS from the latest observation. This happens when the autoregressive prediction of the current embedding (conditioned on the action sequence since the last planning iteration) can not be reidentified with the embedding of the current observation. Moreover, the algorithm stores all observed latent transitions in a lookup table and, when replanning, it only trusts the forward model on previously unseen observation/action pairs. A detailed description can be found in Suppl. C.2.

## 3 Environments

In order to benchmark both perception and abstract reasoning, we empirically show the feasibility of our method on three challenging procedurally generated environments. These include the Maze environment from the procgen suite [13], as well as DigitJump and IceSlider, two combinatorially hard environments which stress the reasoning capabilities of a learning agent, or even of an human player. In the context of our work, the term "combinatorial hardness" is used loosely. We refer to an environment as "combinatorially hard" if only very few of the exponentially many trajectories actually lead to the goal, while deviating from them often results in failure (e.g. DigitJump or IceSlider). Hence, some "intelligent" search algorithm is required. In this way, the process of retrieving a successful policy resembles that of a graph-traversing algorithm. The last two environments are made available in a public repository [1], where they can also be tested interactively. More details on their implementation are included in Suppl. D.

**ProcGenMaze** The ProcgenMaze environment consists of a family of procedurally generated 2D mazes. The agent starts in the bottom left corner of the grid and needs to reach a position marked by a piece of cheese. For each level, an unique shortest solution exists, and its length is usually distributed roughly between 1 and 40 steps. This environment presents significant intra-level variability, with different sizes, textures, and maze structures. While retrieving the optimal solution in this environment is already a non-trivial task, its dynamics are uniform and actions only cause local changes in the observations. Moreover, ProcgenMaze is a forgiving environment in which errors can always be recovered from. In the real world, many operations are irreversible, for instance, cutting/breaking objects, gluing parts, mixing liquids, etc. Environments containing remote controls, for example, show non-local effects. We use these insights to choose the additional environments.

| ProcgenMaze | DigitJump | IceSlider |
|---|---|---|
| 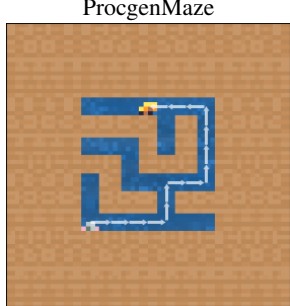 | 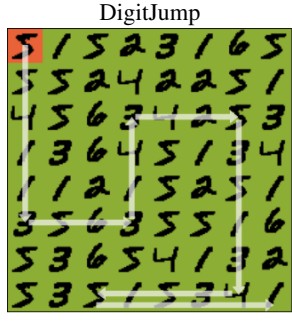 | 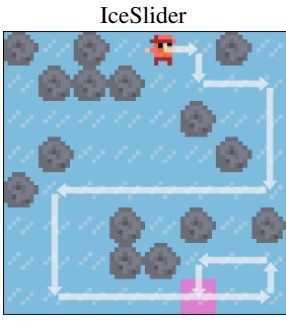 |

Figure 5: Environments. Initial observations and one-shot PPGS's solution (arrows) of a random level of each of the three environments. ProcgenMaze is from [13]. DigitJump and IceSlider are proposed by us and can be accessed at [1].

**IceSlider**    IceSlider is in principle similar to ProcgenMaze, since it also consists of procedurally generated mazes. However, each action propels the agent in a direction until an obstacle (a rock or the borders of the environments) is met. We generate solvable but unforgiving levels that feature irreversible transitions, that, once taken, prevent the agent from ever reaching the goal.

**DigitJump**    DigitJump features a distribution of randomly generated levels which consist of a 2D 8x8 grid of handwritten digits from 1 to 6. The agent needs to go from the top left corner to the bottom right corner. The 4 directional actions are available, but each of them causes the agent to move in that directions by the number of steps expressed by the digit on the starting cell. Therefore, a single action can easily transport the player across the board. This makes navigating the environment very challenging, despite the reduced cardinality of the state space. Moreover, the game presents many cells in which the agent can get irreversibly stuck.

## 4   Related Work

**World Models and Reinforcement Learning**    The idea of learning to model an environment has been widely explored in recent years. Work by Oh et al. [32] and Chiappa et al. [11] has argued that modern machine learning architectures are capable of learning to model the dynamics of a generic environment reasonably well for non-trivial time horizons. The seminal work by Ha and Schmidhuber [17] built upon this by learning a world model in a low-dimensional latent space instead of conditioning predictions on observations. They achieved this by training a VAE on reconstructing observations and a recurrent network for sampling latent trajectories conditioned on an action sequence. Moreover, they showed how sample efficiency could be addressed by recovering a simple controller acting directly on latent representations through an evolutionary approach.

This initial idea was iteratively improved along two main directions. On one hand, some subsequent works focused on learning objectives and suggested to jointly train encoding and dynamics components. Hafner et al. [18] introduced a multi-step variational inference objective to encourage latent representations to be predictive of the future and propagate information through both deterministic and stochastic paths. On the other hand, authors proposed to learn to act in the latent space by using zero-order methods [18] such as CEM [36] or policy gradient techniques [19, 20]. These improvements gradually led to strong model-based RL agents capable of achieving very competitive performance in continuous control tasks [19] and on the Atari Learning Environment [7, 10, 20].

Relying on image reconstruction can however lead to vulnerability to visual noise: to overcome this limitation Okada and Taniguchi [33] and Zhang et al. [43] forgo the decoder network, while the latter proposes to rely on the notion of bisimilarity to learn meaningful representations. Similarly, Gelada et al. [16] only learn to predict rewards and action-conditional state distributions, but only study this task as an additional loss to model-free reinforcement learning methods. Another relevant approach is that of [44], who propose to learn a discrete graph representation of the environment, but their final goal is that of recovering a series of subgoals for model-free RL.

A strong example of how world models can be coupled with classical planners is given by MuZero [38]. MuZero trains a recurrent world model to guide a Monte Carlo tree search by encouraging

hidden states to be predictive of future states and a sparse reward signal. While we adopt a similar framework, we focus on recovering a discrete structure in the latent space in order to reidentify states and lower the complexity of the search procedure. Moreover, we do not rely on reward signals, but only focus on learning the dynamics of the environment.

**Neuro-algorithmic Planning**    In recent years, several other authors have explored the intersection between representation learning and classical algorithms. This is the case, for instance, of Ichter and Pavone [23], Kumar et al. [26], Kuo et al. [27] who rely on sequence models or VAEs to propose trajectories for sampling-based planners. Within planning research, Yonetani et al. [42] introduce a differentiable version of the A* search algorithm that can learn suitable representations from images with supervision. The most relevant line of work to us is perhaps the one that attempts to learn representations that are suitable as an input for classical solvers. Within this area, Asai and Fukunaga [4], Asai and Muise [5] show how symbolic representations can be extracted from complex tasks in an end-to-end fashion and directly fed into off-the-shelf solvers. More recently, Vlastelica et al. [40] frames MDPs as shortest-path problems and trains a convolutional neural network to retrieve the weights of a fixed graph structure. The extracted graph representation can be solved with a combinatorial solver and trained end-to-end by leveraging the blackbox differentiation method [35].

**Visual Goals**    A further direction of relevant research is that of planning to achieve multiple goals [30]. While the most common approaches involve learning a goal-conditioned policy with experience relabeling [3], the recently proposed GLAMOR [34] relies on learning inverse dynamics and retrieves policies through a recurrent network. By doing so, it can achieve visual goals without explicitly modeling a reward function, an approach that is sensibly closer to ours and can serve as a relevant comparison. Another method that sharing a similar setting to ours is LEAP [31], which also attempts to fuse reinforcement learning and planning; however, its approach is fundamentally different and designed for dense rewards and continuous control. Similarly, SPTM [37] pursues a similar direction, but requires exploratory traversals in the current environment, which would be particularly hard to obtain due to procedural generation.

## 5    Experiments

The purpose of the experimental section is to empirically verify the following claims: (i) PPGS is able to solve challenging environments with an underlying combinatorial structure and (ii) PPGS is able to generalize to unseen variations of the environments, even when trained on few levels. We aim to demonstrate that forming complex plans in these simple-looking environments is beyond the reach of the best suited state-of-the-art methods. Our approach, on the other hand, achieves non-trivial performance. With this in mind, we did not insist on perfect fairness of all comparisons, as the different methods have different type of access to the data and the environment. However, the largest disadvantage is arguably given to our own method.

While visual goals could be drawn from a distribution $p(g)$, we evaluate a single goal for each test level matching the environment solution (or the only state that would give a positive reward in a sparse reinforcement working framework). This represents a very challenging task with respect to common visual goal achievement benchmarks [34], while also allowing comparisons with reward-based approaches such as PPO [39]. We mainly evaluate the success rate, which is computed as the proportion of solved levels in a set of 100 unseen levels. A level is considered to be solved when the agent achieves the visual goal (or receives a non-zero reward) within 256 steps.

**Choice of Baselines**    Our method learns to achieve visual goals by planning with a world model learned on a distribution of levels. To the best of our knowledge, no other method in the literature shares these exact settings. For this reason, we select three diverse and strong baselines and we make our best efforts for a fair comparison within our computational limits.

PPO [39] is a strong and scalable policy optimization method that has been applied in procedurally generated environments [13]. While PPGS requires a visual goal to be given, PPO relies on a (sparse) reward signal specializing on a unique goal per level. DreamerV2 [20] is a model-based RL approach that also relies on a reward signal, while GLAMOR [34] is more aligned with PPGS as it is also designed to reach visual goals in absence of a reward.

While we restrict PPGS to only access an offline dataset of low-quality random trajectories, all baselines are allowed to collect data on policy for a much larger number of environment steps. More considerations on these baselines and on the fairness of our comparison can be found in Suppl. B. Furthermore, we also consider a non-learning naive search algorithm (GS ON IMAGES) thoroughly described in C.3.

A comprehensive ablation study of PPGS can be found in Section A of the Appendix.

## 5.1 Comparison of Success Rates

Our first claim is supported by Figure 6. PPGS outperform its baselines across the three environments. The gap with baselines is smaller in ProcgenMaze, a forgiving environment for which accurate plans are not necessary. On the other hand, ProcgenMaze involves long-horizon planning, which can be seen as a limitation to one-shot PPGS. As the combinatorial nature of the environment becomes more important, the gap with all baselines increases drastically.

PPO performs fairly well with simple dynamics and long-term planning, but struggles more when combinatorial reasoning is necessary. GLAMOR and DreamerV2 struggle across the three environments, as they likely fail to generalize across a distribution of levels. The fact that GS ON IMAGES manages to rival other baselines is a testament to the harshness of the environments.

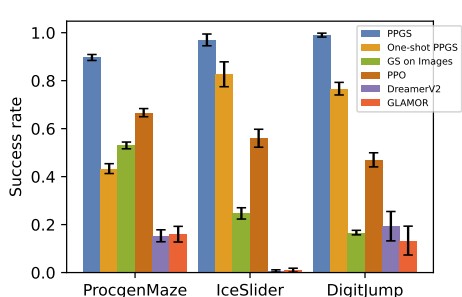

Figure 6: Success rates across the three environments. One-shot planning is competitive with the full method on shorter time horizons.

## 5.2 Analysis of Generalization

The inductive biases represented by the planning algorithm and our training procedure ensure good generalization from a minimal number of training levels. In Fig. 7, we compare solution rates between PPGS and PPO as the number of levels available for training increases. The same metric for larger training level sets is additionally available in Table 3. Our method generally outperforms its baselines across all environments. In ProcgenMaze, PPGS achieves better success rates than PPO after only

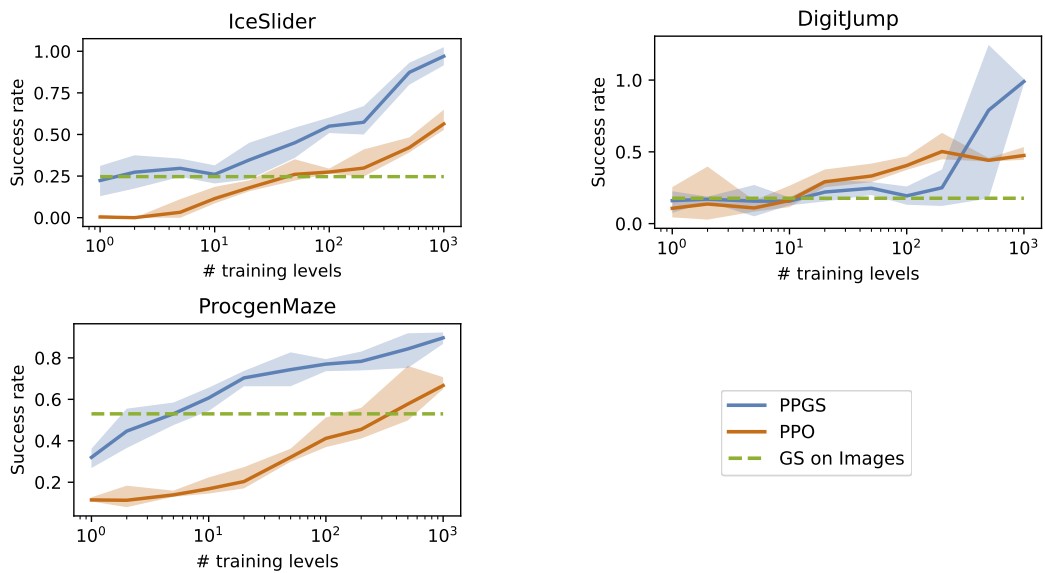

Figure 7: Solution rates of PPGS and PPO as a function of the cardinality of the set of training levels.

seeing two orders of magnitude less level, e.g. 10 levels instead of 1000. Note that PPGS uses only 400k samples from a random policy whereas PPO uses 50M on-policy samples. Due to the harshness of the remaining environments, PPO struggles to find a good policy and its solution rate on unseen levels improves slowly as the number of training levels increases. In IceSlider, PPGS is well above PPO for any size of the training set and a outperforms GS ON IMAGES when only having access to 2 training levels. While having a comparable performance to PPO on small training sets in DigitJump, our method severely outperforms it once approximately 200 levels are available. On the other hand, PPO's ability to generalize plateaus. These results show that PPGS quickly learns to extract meaningful representations that generalize well to unseen scenarios.

## 6    Discussion

**Limitations**    The main limitations of our method regard the assumptions that characterize the class of environments we focus on, namely a slowly expanding state space and discrete actions. In general, due to the complexity of the search algorithms, scaling to very large action sets becomes challenging. Moreover, a single expansion of the search tree requires a forward pass of the dynamics network, which takes a non-negligible amount of time. Finally, the world model is a fundamental component and the accuracy of the forward model is vital to the planner. Training an accurate forward model can be hard when dealing with exceedingly complex observations: very large grid sizes in the environments are a significant obstacle. On the other hand, improvements in the world model would directly benefit the whole pipeline.

**Conclusion**    Hard search from pixels is largely unexplored and unsolved, yet fundamental for future AI. In this paper we presented how powerful graph planners can be combined with learned perception modules to solve challenging environment with a hidden combinatorial nature. In particular, our training procedure and planning algorithm achieve this by (i) leveraging state reidentification to reduce planning complexity and (ii) overcoming the limitation posed by information-dense observations through an hybrid forward model. We validated our proposed method, PPGS, across three challenging environments in which we found state-of-the-art methods to struggle. We believe that our results represent a sensible argument in support of the integration of learning-based approaches and classical solvers.

## Acknowledgments and Disclosure of Funding

We acknowledge the support from the German Federal Ministry of Education and Research (BMBF) through the Tübingen AI Center (FKZ: 01IS18039B). Georg Martius is a member of the Machine Learning Cluster of Excellence, funded by the Deutsche Forschungsgemeinschaft (DFG, German Research Foundation) under Germany's Excellence Strategy – EXC number 2064/1 – Project number 390727645.

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
