# Supplementary Material for:
# Planning from Pixels in Environments
# with Combinatorially Hard Search Spaces

## A    Ablation Study

PPGS relies on crucial architectural choices that we now set out to motivate. We do so by performing an ablation study and questioning each of the choices individually to show its contribution to the final performance.

**World Model**    We evaluate the impact of different choices on the world model by retraining it from scratch and reporting the success rate of the full planner in Table 1. We also compute two latent metrics, which are commonly used to benchmark latent predictions [25], see below.

| | ProcgenMaze / DigitJump | | | | |
| --- | --- | --- | --- | --- | --- |
| | Success % | H@1 | H@10 | MMR@1 | MMR@10 |
| Our method | 0.91/1.00 | 1.00/1.00 | 0.92/0.99 | 1.00/1.00 | 0.94/1.00 |
| without inverse model | 0.38/0.23 | 1.00/1.00 | 0.77/0.23 | 1.00/1.00 | 0.81/0.33 |
| with fully latent forward model | 0.80/0.34 | 0.98/1.00 | 0.73/0.53 | 0.98/1.00 | 0.81/0.63 |
| without lookup table | 0.39/0.92 | - | - | - | - |
| one-shot, with reidentification | 0.43/0.76 | - | - | - | - |
| one-shot, without reidentification | 0.27/0.31 | - | - | - | - |

Table 1: Ablations. We evaluate the success rate on two environments when removing important components of our world model and planner. For the world model modifications, we also report metrics for predictive accuracy explained in the text. All results are averaged over 3 seeds.

In particular, given a planning horizon $K$, we first collect a random trajectory $(s_t, a_t)_{t=1...L}$ of length $L = 20$ and extract latent embeddings $\{z_t\}_{t=1,...L}$ through the encoder $h_\theta$. We then autoregressively estimate the embedding $z_{K+1}$ using only the first embedding $z_1$ and the action sequence $(a_t)_{t=1...K}$, obtaining a prediction $\hat{z}_{K+1}$.

We repeat this process for $N$ trajectories, obtaining $N$ sequences of latent embeddings $(z_t^n)_{t=1...L}^{n=1...N}$ and $N$ predictions $\{\hat{z}_{K+1}^n\}^{n=1...N}$. We compute $\text{rank}(\hat{z}_{K+1}^n)$ as the lowest $k$ such that $\hat{z}_{K+1}^n$ is in the $k$-nearest neighbors of $z_{K+1}^n$ considering all other embeddings $\{z_t^n\}_{t=1,...L}$. We can finally compute $H@K = \frac{1}{N} \sum_{n=1}^{N} \mathbf{1}_{\text{rank}(\hat{z}_{K+1}^n)=1}$ and $\text{MMR@K} = \frac{1}{N} \sum_{n=1}^{N} \frac{1}{\text{rank}(\hat{z}_{K+1}^n)}$.

We found that training an inverse model is crucial for learning good representations. Even if the uninformative loss $L_{\text{margin}}$ introduced in Equation 4 already helps with avoiding the collapse in the latent space, we were not successful in training the forward model to high predictive accuracy unless the inverse model was jointly trained, despite further hyperparameter tuning. We can hypothesize that the inverse model enforces a regular structure in the latent space, which is in turn helpful for training the rest of the world model.

On the other hand, we find that the contribution of an hybrid forward model is more environment-dependent. We ablate this by introducing a forward model that only takes the state embedding $z_t$ and an action $a_t$ as input to predict the next embedding $z_{t+1}$, without having access to a contextual observation $s_c$. In this case, the forward model can be implemented as a layer-normalized MLP with 3 layers of 256 neurons. When adopting this fully latent forward model, predictive accuracy drops sensibly. While success rate also drops sharply in DigitJump, this is not the case for ProcgenMaze. We believe that this can be motivated by the fact that several levels of ProcgenMaze could be solved in few steps, only knowing the local structure of the maze with respect to the starting position of the agent. In DigitJump, on the other hand, the agent can easily move across the environment and needs global information to plan accurately.

**Planner**   Evaluating the planning algorithm does not require retraining the world model. We first show the importance of the lookup table. Without correcting the world model's inaccuracies, the planner is not able to recover from incorrect trajectories. As a result, the success rate is comparable to that of one-shot planning. Finally, we empirically validate the importance of state reidentification: when disabled, the BFS procedure is forced to randomly discard new vertices due to the exponential increase in the size of the graph. Because of this, promising trajectories cannot be expanded and the planner is only effective on simple levels which only require short-term planning.

**Failure Cases**   We now present a visual rendition of failure cases for one-shot PPGS in Fig. 8, together with the correct policy retrieved by the full planner.

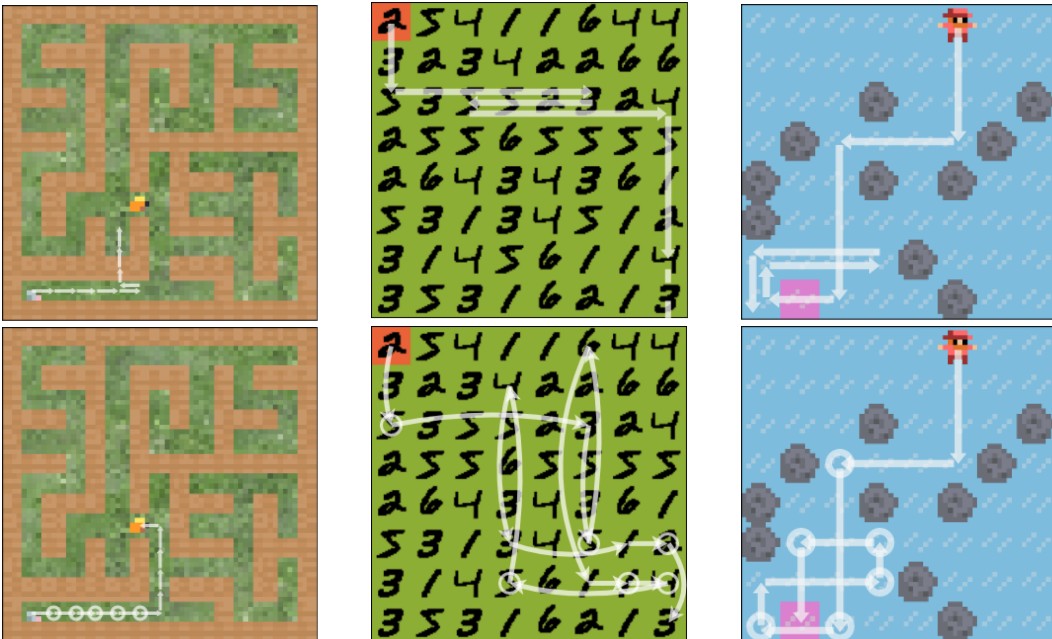

Figure 8: Failure cases. On the first row, a level from each environment that one-shot PPGS fails to solve (the white arrows represent the policy). On the second row, the policies corrected by a full planner, which is able to solve all levels. A white circle is drawn when PPGS recomputes its policy.

**Iterative Model Improvement**   In general settings, collecting training trajectories by sampling actions uniformly at random does not grant sufficient coverage of the state space. In this case, the planners designed for PPGS can be deployed for data collection while training the world model. To show the potential of this approach in our current setting, we train our method starting from a small number of random trajectories and iteratively collecting on-policy data. We compare the performance of one-shot planning on IceSlider when trained on 1k levels. We measure success rates on unseen levels for (A) the default setting (400k random training samples), (B) in a low-data scenario (100k random training samples), and (C) when iteratively adding on-policy transitions to a small initial training set of 100k random training samples. In this last case, we collect 100k additional on-policy samples every 5 epochs. At 20 epochs we observe that on-policy data collection is able to accelerate and stabilize learning (see Table 2). When training is completed (40 epochs), (A) and (C) reach the same performance, while (B) does not improve. We believe that this hints that

- random trajectories are enough to eventually cover the state space of IceSlider well and,

- the planner can be effectively used for data collection and iterative improvements.

We therefore believe that collecting data on-policy in an RL-like loop is crucial in environments requiring more exploration and represents an interesting direction.

| | **IceSlider** - One-shot success % on unseen levels | | |
|---|---|---|---|
| | (A) 400k offline | (B) 100k offline | (C) 100k off + 300k on-policy |
| 20 epochs | 0.331±0.27 | 0.040±0.01 | **0.778±0.07** |
| 40 epochs | 0.826±0.05 | 0.043±0.01 | **0.857±0.04** |

Table 2: Performance when training with datasets of various sizes and with on-policy data collection. Collecting trajectories generated by the planner can accelerate learning.

# B  Choice of Baselines and Fairness

After introducing the fundamental reasons behind our choice of baselines in Sec. 5, we present our reasoning and experimental setup with respect to each of the methods.

**PPO**    PPO [39] is a strong model-free RL algorithm. Unlike PPGS, PPO requires a reward signal instead of visual goals. We grant PPO an advantageous setting by allowing on-policy data collection for 50M environment steps, which is in stark contrast to the offline dataset of 400k random transitions that PPGS is trained on. We use the implementation and hyperparameters presented by Cobbe et al. [13] for the Procgen suite, due to its similarity to the rest of the environments. While PPGS is tuned on ProcgenMaze and keeps its hyperparameters fixed across environments, we favor PPO by tuning the number of timesteps per rollout according to the environment to account for the possibility of getting stuck in a funnel state.

**GLAMOR**    GLAMOR [34] learns inverse dynamics to achieve visual goals in Atari games. Similarly to PPGS, GLAMOR does not require a reward signal but needs to receive a visual goal. The only difference with PPGS in terms of settings is that we allow GLAMOR to collect data on-policy and for more interactions (2M). At evaluation time we deploy a strictly more forgiving scheme for GLAMOR, which is described in the original paper [34]. As GLAMOR is designed to approximately reach its goals, we also accept trajectories that terminate near the actual goal as viable solutions. Hyperparameters for GLAMOR were tuned by the original authors in Atari, which is a visually comparable setting.

**DreamerV2**    DreamerV2 [20] is a model-based RL approach reaching state-of-the-art performance in discrete games and continuous control domains. We use the original implementation for Atari environments. Due to its large computational requirements, we are only able to run DreamerV2 for a reduced number of steps, totaling 4M. We remark that while this is not enough for performance on Atari to converge, it is shown by the original authors to be sufficient for solving a significant number of games.

All cited codebases that we use are publicly available under a MIT license.

# C  Further Implementation Details

In this section we report relevant implementation choices for PPGS. In any case, we refer to our code [2] for precise details.

## C.1  World Model

**Encoder.**    The encoding function $h_\theta$ is learned by a convolutional neural network. The output of the convolutional backbone of a ResNet-18 [21] is fed through a single fully connected layer with $d$ units, where $d = 16$ is the size of the latent space $Z$. The output of the network is normalized to unit L2 norm.

**Forward Model.**    From an architectural perspective, our hybrid forward model transforms the state embedding $z_t$ through a deconvolutional network and concatenates it to the RGB observation $s_c$ and a batchwise one-hot tensor representing the action. The result is processed through a second ResNet-18 to predict the next embedding. We found it to be irrelevant whether to train the network to predict a state representation $z_{t+1}$ or a latent offset $z_{t+1} - z_t$: for our experiments we choose the former.

Similarly to Hafner et al. [19] we find that, in practice, explicitly encouraging representations to be predictive for longer horizons (for instance through a multi-step loss) does not appear to be helpful. For this reason, we only train for one-step predictions, as noted in Equation 2.

**Inverse Model.** To enforce a simpler structure in the latent space, we implement the inverse model $p_\omega$ as a low-capacity one-layer MLP with ReLU activations, 32 neurons and layer normalization [6].

**Hyperparameters** Hyperparameters are fixed in all experiments unless explicitly mentioned. More importantly, we deploy the same set of hyperparameters across all three environments, after tuning them on ProcgenMaze via grid search. The latent margin $\varepsilon$ is set to 0.1 and the dimensionality of the latent space $d$ is set to 16. The world model we propose is optimized using Adam [24] with learning rate $\lambda = 0.001$ for all components and parameters $\varepsilon = 0.00001, \beta_1 = 0.9, \beta_2 = 0.999$. All components are trained for 40 epochs with a batch size of 128. The losses are combined as in Equation 5 with weights $\alpha = 10, \beta = 1$, although our architecture shows robustness to this choice. Training the world model takes approximately 20 hours on a single NVIDIA ampere GPU.

## C.2 Planner

In this subsection, we present both planners (one-shot and full) more in detail.

**One-shot Planner** Algorithm 2 offers a more thorough description of the one-shot planner introduced in Algorithm 1. Given a visual goal $g$ and an initial observed state $s$, maximizing discounted rewards corresponds to recovering the shortest action sequence $(a_i)_{1,...,n}$ such that $s_n = g$.

For this purpose, Algorithm 2 efficiently builds and searches the latent graph. It has access to an initial high-dimensional state $s$ and a visual goal $g$; it keeps track of a list of visited vertices $V$ and of a set of leaf vertices $L$. For each visited latent embedding, the algorithm stores the action sequence leading to it in a dictionary $D$. A key part of the algorithm is represented by the `filter` function. The `filter` function receives as input the new set of leaves $L'$, from which vertices reidentifiable with visited states have already been discarded. The function removes elements of $L'$ until no pair of states is too close. This is done by building a secondary graph with the elements of $L'$ as vertices and edges between all pairs of vertices at less than $\frac{\epsilon}{2}$ distance. A set of non-conflicting elements can then be recovered by approximately solving a minimum vertex cover problem. If the state space of the environment grows exponentially with the planning horizon, or if the world model fails to reidentify bisimilar states, $L'$ can still reach impractically large sizes. For this reason, after resolving conflicts, if its cardinality is larger than a cutoff $C = 256$, $|L_{t+1}| - C$ elements are uniformly sampled and removed.

**Full Planner** The full planner used by PPGS introduces the possibility of online replanning in an MPC approach. It autoregressively computes a latent trajectory $T$ conditioned on the action sequence $P$ retrieved by one-shot planning. At each step, the current observation is projected to the latent space to check if it can be reidentified with the predicted embedding in $T$. When this is not possible, the action sequence $P$ is recomputed. Moreover, the planner gradually fills a latent transition buffer $B$. Forward predictions are then computed according to $\hat{f}_\theta(z, a, s)$, which returns $z'$ if $(z, z', a) \in B$, otherwise it queries the learned forward model $f_\theta$. As a side note, when replanning while using the full planner, the planning horizon $T_{\max}$ is set to 10 steps. We report the method in full in Algorithm 3.

## C.3 GS on Images

Our baselines include a graph search algorithm in observation space that does not involve any learned component. We refer to this algorithm as GS ON IMAGES and it can be seen as a measure of how hard an environment is when relying on reconstructing the state diagram to solve it. GS ON IMAGES assumes solely on the deterministic nature of the environment. Given a starting state $s$, a goal state $g$ and the action set $A$, GS ON IMAGES plans as shown in Algorithm 4. It relies on a dictionary $A_{\text{left}}$ which stores, for each visited state, the set of actions that have not been attempted yet, and on a graph representation of the environment $G = (V, E)$, where $V$ is the set of visited states and $E$ contains the observed transitions between states and labeled by an action.

As a side note on this algorithm's performance, we remark that, unlike the remaining methods, it strongly depends on the absence of visual noise and distractors. In particular, this method is bound to

---

**Algorithm 2** One-shot PPGS

---

**Input:** $s, g$
**Output:** action sequence $(a_i)_{1,\ldots,n}$

1:   $z, z_g = h_\theta(s_1), h_\theta(g)$                                                       $\triangleright$ project to latent space
2:   $V = L = \{z\}$
3:   $D = \{z : []\}$
4:   **while** for $T_{\text{MAX}}$ steps **do**
5:      $L' = \emptyset$
6:      **for** $z \in L$ **do**                                               $\triangleright$ grow the latent graph
7:          **for** $a \in A$ **do**
8:              $z' = f_\phi(z, a, s)$
9:              **if** $\min_{v \in V} \|z' - v\|_2 > \frac{\varepsilon}{2}$ **then**             $\triangleright$ skip if already visited
10:                  $L' = L' \cup \{z'\}$
11:                  $D[z'] = D[z] + [a]$
12:              **end if**
13:          **end for**
14:      **end for**
15:      $L = \texttt{filter}(L')$      $\triangleright$ select the largest group of elements such that no pair is too close
16:      $z^\star = \arg\min_{z \in L} \|z - z_g\|_2$
17:      **if** $\|z^\star - z_g\|_2 \leq \frac{\varepsilon}{2}$ **then**                   $\triangleright$ if $z^\star$ can be reidentified with the goal
18:          return $D[z^\star]$
19:      **end if**
20:      $V = V \cup L$                                         $\triangleright$ add leaves to visited set
21: **end while**

---

fail in environments in which this assumption does not hold: visual noise would render reidentification meaningless for GS on Images and the baseline would not be able to avoid revisiting vertices of the latent graph.

### C.4 Data Collection

As our method solely relies on accurately modeling the dynamics of the environment, the only requirement for training data is sufficient coverage of the state space. In most cases, this is satisfied by collecting trajectories offline according to a uniformly random policy. The ability to leverage a fixed dataset of trajectories draws PPGS closer to off-policy methods or even batch reinforcement learning approaches.

In practice, unless specified otherwise, we collect 20 trajectories of 20 steps from a set of $n = 1000$ training levels, for a total of $400k$ environment steps. One exception is made for ProcgenMaze, for which we also set a random starting position at each episode, since uncorrelated exploration is not sufficient to cover significant parts of the state.

## D   Environments

In this section, we present a few remarks on the environments chosen. For ProcgenMaze, we choose what is reported as the *easy* distribution in Cobbe et al. [13]. This corresponds to grids of size $n \times n$, with $3 \leq n \leq 15$; each cell of the grid is either a piece of wall or a corridor. In IceSlider, the agent always starts in the top row and needs to descend to a goal on the bottom row. It is not sufficient to slide over the goal, but the agent needs to come to a full stop on the correct square. In DigitJump, the handwritten digits are the same across training and test levels. Their frequency and position does of course change.

All environments return observations as 64x64 RGB images. ProcgenMaze and IceSlider are rendered in a similar style to ATARI games, while the DigitJump is a grid of MNIST [28] digits that highlights the cell at which the agent is positioned. The action space in all cases is restricted to four cardinal actions (UP, DOWN, LEFT, RIGHT) and a no-op action, for a total of 5 actions.

**Algorithm 3** PPGS (full planner)

---

**Input:** $s, g$

1: $z = h_\theta(s)$
2: $B = \emptyset$        ▷ set of observed latent transitions
3: $P = \texttt{one\_shot\_PPGS}(s, g)$        ▷ retrieve initial policy
4: $T = [z]$
5: **for** a in P **do**:
6:      $T = T + [\hat{f}_\phi(T[-1], a, s)]$        ▷ autoregressively predict latent trajectory T
7: **end for**
8: $T.\text{pop}(0)$        ▷ discard first embedding
9: **while** $s \neq g$ **do**
10:      $a = P.\text{pop}(0)$        ▷ take first action and remove it from the action list
11:      take action $a$ and reach state $s'$
12:      $z' = h_\theta(s')$
13:      $B = B \cup \{(z, z', a)\}$
14:      $z_{\text{pred}} = T.\text{pop}(0)$        ▷ retrieve predicted embedding
15:      **if** $\|z_{\text{pred}} - z'\|_2 > \frac{\varepsilon}{2}$ or A = [] **then**        ▷ if the latent trajectories does not match predictions
16:          $A = \texttt{one\_shot\_PPGS}(s', g)$        ▷ replan
17:          $T = [z]$
18:          **for** a in P **do**:
19:             $T = T + [\hat{f}_\phi(T[-1], a, s)]$        ▷ autoregressively predict latent trajectory T
20:          **end for**
21:          $T.\text{pop}(0)$        ▷ discard first embedding
22:      **end if**
23:      $s = s'$
24:      $z = z'$
25: **end while**

---

Examples of expert trajectories are shown in Fig. 9. For more information on the environments, we refer the reader to our code [1].

## E    Numerical Results

We finally include the full numerical results from Fig. 6 and Fig. 7 in Table 3.

| **ProcgenMaze** - Success % on unseen levels when training on $n$ levels | | | | | | | | | | | | |
|---|---|---|---|---|---|---|---|---|---|---|---|---|
| | $n$=1 | 2 | 5 | 10 | 20 | 50 | 100 | 200 | 500 | 1000 | 2000 | 5000 | 10000 |
| PPGS | 0.320 | 0.447 | 0.530 | 0.607 | 0.703 | 0.743 | 0.770 | 0.783 | 0.843 | 0.897 | 0.850 | 0.880 | 0.880 |
| PPO | 0.115 | 0.113 | 0.139 | 0.168 | 0.203 | 0.321 | 0.417 | 0.455 | 0.577 | 0.667 | 0.746 | 0.853 | 0.880 |
| DreamerV2 | - | - | - | - | - | - | - | - | - | 0.153 | - | - | - |
| GLAMOR | - | - | - | - | - | - | - | - | - | 0.100 | - | - | - |

| **IceSlider** - Success % on unseen levels when training on $n$ levels | | | | | | | | | | | | |
|---|---|---|---|---|---|---|---|---|---|---|---|---|---|
| | $n$=1 | 2 | 5 | 10 | 20 | 50 | 100 | 200 | 500 | 1000 | 2000 | 5000 | 10000 |
| PPGS | 0.223 | 0.273 | 0.296 | 0.260 | 0.347 | 0.450 | 0.550 | 0.573 | 0.873 | 0.970 | 0.960 | 0.965 | 0.965 |
| PPO | 0.004 | 0.000 | 0.032 | 0.115 | 0.179 | 0.260 | 0.274 | 0.298 | 0.421 | 0.564 | 0.565 | 0.590 | 0.601 |
| DreamerV2 | - | - | - | - | - | - | - | - | - | 0.007 | - | - | - |
| GLAMOR | - | - | - | - | - | - | - | - | - | 0.010 | - | - | - |

| **DigitJump** - Success % on unseen levels when training on $n$ levels | | | | | | | | | | | | |
|---|---|---|---|---|---|---|---|---|---|---|---|---|---|
| | $n$=1 | 2 | 5 | 10 | 20 | 50 | 100 | 200 | 500 | 1000 | 2000 | 5000 | 10000 |
| PPGS | 0.160 | 0.170 | 0.157 | 0.157 | 0.220 | 0.247 | 0.193 | 0.250 | 0.790 | 0.990 | 0.960 | 0.940 | 0.980 |
| PPO | 0.107 | 0.137 | 0.109 | 0.161 | 0.291 | 0.331 | 0.405 | 0.502 | 0.441 | 0.475 | 0.513 | 0.488 | 0.492 |
| DreamerV2 | - | - | - | - | - | - | - | - | - | 0.193 | - | - | - |
| GLAMOR | - | - | - | - | - | - | - | - | - | 0.133 | - | - | - |

Table 3: Generalization results. This table presents the numerical results used to produce Fig. 6 and Fig. 7. All metrics are averaged over 3 random seeds.

**Algorithm 4** GS ON IMAGES

**Input:** $s, g$

1: $A_{\text{left}} = \{s : A\}$
2: $V = \{s\}$
3: $E = \emptyset$
4: **while** $s \neq g$ **do**
5:     **if** $A_{\text{left}}[s] = \emptyset$ **then**
6:         **if** $\exists s' \in V$ such that $A_{\text{left}}[s'] \neq \emptyset$ and $s'$ is reachable from $s$ **then**
7:             find and apply action sequence to reach closest $s' \in V$
8:             $s = s'$
9:         **else**
10:             **return**
11:         **end if**
12:     **else**
13:         $a \sim \mathcal{U}(A_{\text{left}}[s])$                          $\triangleright$ uniformly sample among remaining actions
14:         $A_{\text{left}}[s] = A_{\text{left}}[s] \setminus \{a\}$
15:         take action $a$ and reach state $s'$
16:         **if** $s' \notin V$ **then**
17:             $V = V \cup \{s'\}$
18:             $A_{\text{left}}[s] = A$
19:         **end if**
20:         $E = E \cup (s, s', a)$
21:     **end if**
22: **end while**

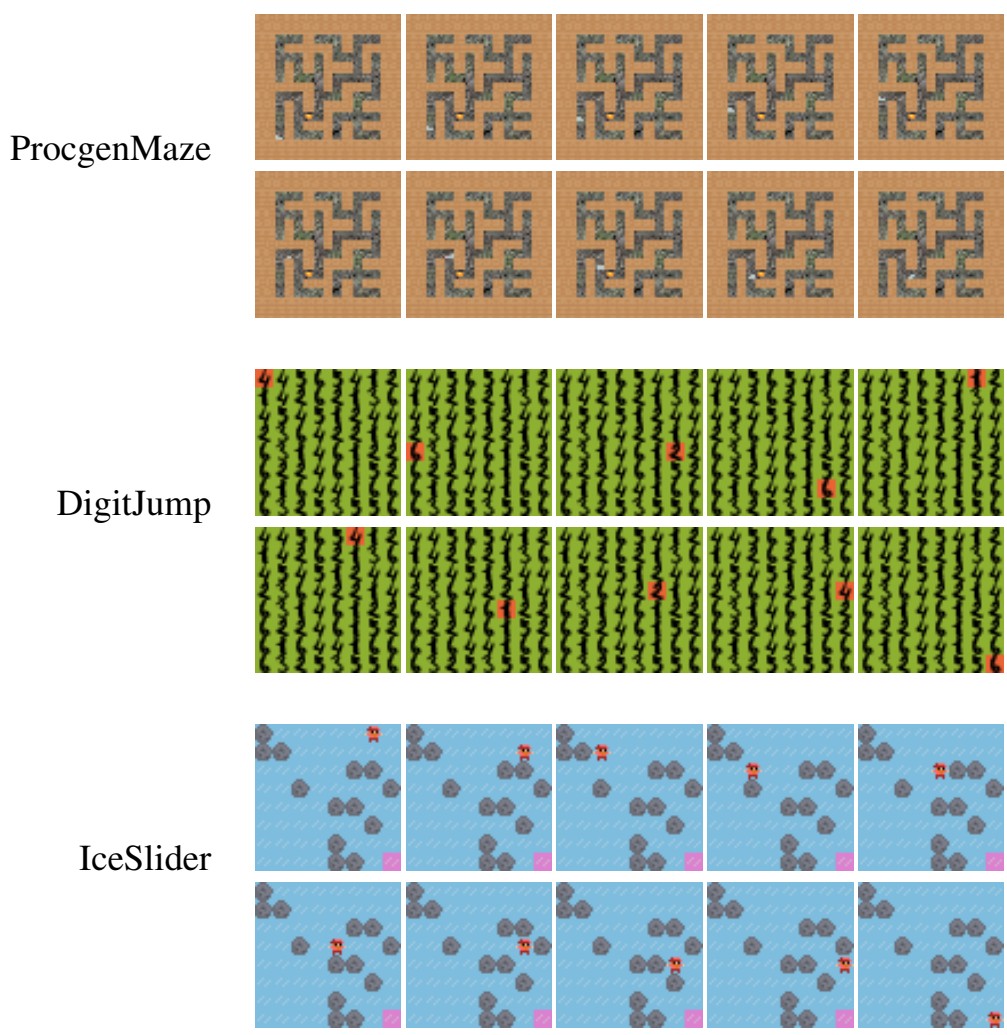

Figure 9: Expert trajectories for a level extracted from each of the environments.