# OpenReview forum: "Planning from Pixels in Environments with Combinatorially Hard Search Spaces"
_NeurIPS.cc/2021/Conference — NeurIPS 2021 Poster_

### Official Review · Reviewer_5kFP · 2021-07-14

**Rating:** 6
**Confidence:** 3

**Summary:**

This paper introduces a planning method for solving combinatorial environments from pixels called PPGS. PPGS learns a latent space using a ResNet encoder which is trained such that a forward model and inverse model can be learned using the representation. Finally, it is constrained to encourage margins in the latent space. During goal-conditioned planning, visual inputs are mapped into the latent space, discretized, and a BFS-like search algorithm is performed. PPGS achieves strong empirical results on the Procgen maze environment and two 2D Atari-like environments which the authors newly introduce in the paper.

**Limitations And Societal Impact:**

I was unable to find much discussion in the paper specifically about the limitations of PPGS -- it would be great if the authors could add some discussion. However, I believe that one major limitation as mentioned earlier is that the success rate of a random policy must reach a minimum threshold to be able to build a dataset to learn world models which can be used for planning without any additional data collection.

**Main Review:**

Originality: The choices of components in the objective used to learn the latent space are well-motivated and seem sensible. While the idea of planning from pixels using a learned latent space is not novel, to my knowledge the training objective used in this paper contains new elements of combining forward and inverse model training, as well as a discretization step to allow planning on a graph.

Quality: The method is not compared to other model-based planning methods which are goal-conditioned and do not require a reward signal, for example, SPTM (Savinov 2018). These may be expected to perform better than methods which require rewards due to generalization, better than GLAMOR by performing explicit planning, and better than “GS on images” due to being goal-directed.

In motivating the development of new environments, it’s stated “its dynamics are uniform and actions only cause local changes in the observations. Moreover, ProcgenMaze is a forgiving environments in which errors can always be recovered from. We use these insights to choose the additional environments”. While this seems true, the choice of making new evaluation environments more difficult in these axes (nonlocal changes from actions and irrecoverable errors) could be better justified for example in terms of downstream application.

The relatively weak performance of Dreamerv2 and GLAMOR is explained in the text by lack of generalization across a variety of levels. Can this be verified experimentally, e.g. by looking at the performance of these algorithms on fixed levels?

How sensitive is the method to the choice of $\epsilon$ in Equation 4, and how important is adding the margin loss to performance? Was this value tuned prior to experiments? The ablations do not include an analysis of this aspect.

Clarity: It seems like PPGS can only work if planning towards states which have already been seen in the training data or which have neighboring states which have been seen in the training data. This is confusing to me because it seems that the environments are quite difficult, yet this means the random policy must achieve a reasonable success rate on them to collect data in that range. What is the success rate of the uniformly random policy on each of these three environments? Can state coverage be guaranteed for other environments?

Nitpicks:
In figure 2, there is a L_hinge which is not described in the text -- did you mean L_margin?.

Typos: L202: “wepsite” -> “website”, L340: “state-of-the-art method” -> “methods”, L210 “envrionments” -> “environment”

Consider citing:
“Semi-parametric Topological Memory for Navigation”, Savinov et. al 2018

The code for the environments as well as the algorithm are provided and the authors have made efforts to provide details for reproducibility.

Significance: The empirical results are quite impressive -- the environments in the experiments are quite difficult to solve even for humans, and PPGS achieves an impressive success rate on these tasks. Furthermore, the method is demonstrated only using offline trajectories and does not use any further online rollouts. The experimental environments are a nice contribution and the demo/presentation is quite helpful for understanding their complexity.

Recommendation:

I believe that this work is a nice contribution towards planning from visual observations. While I have concerns about necessary conditions for environments it can perform well on, the generalization performance and state representation learning component of the work seems straightforward and effective.

**Time Spent Reviewing:**

3

---

> ### Author Response · Authors · 2021-08-10
> **Response to 5kFP**
>
> We are thankful for your review and the extensive feedback.
>
> ## SPTM
> Thanks for mentioning this method. SPTM pursues a similar direction to our work and we will include it in the related works. Training and deploying SPTM in our setting is problematic as it requires an exploratory sequence/traversal to construct the particular graph for the current level used for planning. In this, our GS baseline is similar to SPTM with an on-the-fly traversal and without a learned similarity metric.  In our settings, such a demonstration is not available. Sadly, we could not implement it as a privileged baseline due to time constraints. A discussion of other baselines is also included in the answer to reviewer XKCd.
>
> ## Motivation for New Environments
> In the real world, many operations are irreversible, for instance, cutting/breaking objects, gluing parts, mixing liquids, etc. Environments containing remote controls, for instance, show non-local effects. We think that these properties are important and currently not well captured in the available environments. We hope that this is the motivation you hinted at and we added it to the manuscript.
>
> ## Baselines on Single Levels
> We experimentally validated this issue. Glamor and Dreamer are able to overfit to a single level, as long as the level is not too complex (i.e. uniform exploration can consistently reach the goal state). Similar behavior can be observed in PPO: when trained on a single level, its success rate either converges to 100% or remains at 0%. On the other hand, when training on a small set of levels, an implicit curriculum can help PPO achieve higher average success rates, as reported in [1].
>
> ## Margin Loss
> We found $L_{margin}$ to play a vital part in our method. It helps prevent the latent space from collapsing. It also enforces a latent margin that is consistent across the latent space and that is fundamental for the planner. Without the $L_{margin}$, this margin would have to be estimated and might be different for different pairs of states. We use the same value for the margin-loss as we use in the reidentification during planning. $\epsilon$ was fixed apriori to 0.1, which we determined through a small grid search, but its choice was not critical. The answer to reviewer FT4Z (see Ablating $L_{margin}$) also contains an ablation and more details on this topic.
>
> ## PPGS on Unseen States
> PPGS is tested on unseen levels with a new topology. In fact, PPGS does not need to have observed similar states at training time. We believe that our method achieves this by learning the underlying dynamics of the environment, which are consistent across levels, at training time. At test time, generalization only requires recovering the topology of the level, which can be done from a single observation.
>
> ## Success of a Random Policy
> We report the success rate of a random policy (averaged over 3 seeds) across all environments in the table below and add it to the manuscript.
> ```
> |        	     |  ProcgenMaze  |   IceSlider   |   DigitJump   |
> | Random Policy   | 0.31 +- 0.008 | 0.15 +- 0.014 | 0.16 +- 0.049 |
> ```
>
> ## Limitations
> We will add the following discussion about the limitations, which felt short due to the page limit in the initial submission.
> The main limitations of our method regard the assumptions that characterize the class of environments we focus on, namely a slowly expanding state space and discrete actions.
> In general, due to the complexity of the search algorithms, scaling to very large action sets becomes challenging. Moreover, a single expansion of the search tree requires a forward pass of the dynamics network, which takes a non-negligible amount of time.
> Finally, the world model is a fundamental component and the accuracy of the forward model is vital to the planner. Training an accurate forward model can be hard when dealing with exceedingly complex observations: very large grid sizes in the environments are a significant obstacle. On the other hand, improvements in the world model would directly benefit the whole pipeline.
>
> [1] Leveraging Procedural Generation to Benchmark Reinforcement Learning, https://arxiv.org/abs/1912.01588

---

### Official Review · Reviewer_Ft4Z · 2021-07-16

**Rating:** 6
**Confidence:** 3

**Summary:**

This paper proposes a model-based planning method for pixel-based control in difficult-to-plan environments in which the full environment is observed through a high-dimensional image. The method proceeds by proposing a particular learning objective for learning image representations that can be used for localization ("reidentification") and planning. After this encoder is learned, the representations are used inside a graph-based planner that observes the current image and target goal image and uses receding-horizon control ("Full Planner") to control the agent. Experiments on an existing set of difficult environments, as well as two newly-proposed environments that satisfy the desired properties (difficulty and observability) compare the method to model-based baselines, a model-free baseline, and a simple learning-free planning baseline. These experiments show the method outperforming the baselines in all key settings, and exhibiting better performance relative to a model-free approach in low-sample regimes.

**Limitations And Societal Impact:**

The paper didn't really discuss broader limitations or broadly characterize failure cases -- the Checklist 1(b) answer points us to sections that narrowly reference performance issues. One of the main limitations is the observability assumption, furthermore, determinism in the dynamics seems quite limiting. These should be discussed somewhere. An inclusion of potential broader impacts was added in Checklist response 1(c), but it should be included in the appendix (?)

**Main Review:**

The setting is an interesting and highly-relevant: model-based control from pixels in a particular class of difficult environments. The proposed method is straightforward (fairly intuitive) and performs well. The proposed evaluation does a reasonable job of highlighting the strengths of the method and deficiencies of existing methods in the setting. I have a few concerns about the problem setup, importance of training objective components, and experiments that I think are important (hence, my initial rating of 6 instead of 7), but I believe should be straightforward to address.

**Problem setup:**
- L60 I think the set of MDPs need the further assumptions that each $|s_i| = |s_j| \forall i\neq j$, and similarly for actions, otherwise the model will need to be able to handle states and actions of different dimensionalities. Furthermore, I think that the problem assumption might match that of Block MDPs [A] -- which assume that each observation contains enough information to uniquely identify the state and implies the existing of a perfect decoder. If true, this would make the problem setup in this paper clearer, in which each $z_t=f(o_t)$ is meant to approximate a true underlying $s_t$. Particularly, it's peculiar to think of the high-dimensional images as Markovian states; the block-MDP assumption fixes this peculiarity.
- The notion of "combinatorial hardness" is important to the setting in which the algorithm is meant to function well, yet it is never precisely defined, which also means that the environments employed are not conclusively shown to possess combinatorial hardness. I think this needs to be formalized in order for the paper to make its claims precise enough.

**Evaluation concerns:**
- Does the margin loss actually contribute to learning a better latent space? Discretization for the graph search could still be performed using a model trained without the margin loss. It would be good to have evidence to show that the margin loss results in improved planning performance. This evaluation could be added to Table 1 in the appendix, and referenced in the main text when the margin loss is discussed. If the following ends up being true, that could look something like "we found the margin loss to result in representations better-suited to our planning method, see Appendix Table 1." Similar ablations are missing for the $\mathcal L_{\text{FW}}$ and $\mathcal L_{\text{CE}}$ losses.
- Success rate fails to disentangle the efficiency with which approaches achieve the goals. I think the paper would have a more robust evaluation if it included the success weighted by path length metric -- see e.g. Eq. 1 from [B]. For instance, it could be true that PPGS finds very efficient paths relative to the baselines, and the baselines merely do a reasonable job of efficiently exploring the space, rather than exhibiting particularly efficient directed behavior.
- L318 "GS on Images manages to rival other baselines is a testament to the harshness of the environments" I'm not sure there's enough evidence to conclude this. I'd say that GS on Images is performing reasonably well (>40% success in all environments). Given that GS on Images is a "simple" baseline that doesn't employ learning, this suggests to me that the environments are not particularly harsh (or they are well-suited to GS on Images), and that the other baselines are merely underperforming / are not well-suited to the task.
- If computationally feasible, it would be good to extend the X axes of Fig 7 IceSlider and ProcgenMaze (and DigitJump), in order to determine if/when the performance of PPGS and PPO overlap. The trends suggest that PPO may reach PPGS performance at ~$10^4$ training levels. Presumably they would eventually overlap. This evaluation would strengthen our understanding of the performance of the approaches as a function of the available training levels.
- L325 "Note that PPGS uses only 400k samples from a random policy whereas PPO uses 50M on-policy samples" This point is too easy to miss. It's not otherwise clear that the amount of training data available to each approach is different. Isn't the 400k samples only true at one of the x-values in Fig7? The number of training samples available to each approach should be made clear in Fig 7. For example, having stacked x-axes for each method in Fig 7 could work, if the number of training examples available to PPGS and PPO are linear functions of the number of levels (current x-axis).

**Minor comments:**
- L60 There's an extra comma in the set; also, (pedantry to follow) include the tuple parens: $\{(S, A, T, G, R, \gamma)_i\}_{i=1}^n$.
- L280 "the largest disadvantage is arguably given to our own method." More explanation is needed. Is this due to the reward-free learning?
- typo: Eq 6 has an extra parenthesis after $z_h^l$.
- L202 typo "wepsite"
- Alg 1 is missing the pretrained encoder as input.

[A] Provably efficient RL with Rich Observations via Latent State Decoding; ICML 2019 http://proceedings.mlr.press/v97/du19b/du19b.pdf

[B] On Evaluation of Embodied Navigation Agents; https://arxiv.org/pdf/1807.06757.pdf



**Time Spent Reviewing:**

3

---

> ### Author Response · Authors · 2021-08-10
> **Reponse to Ft4Z**
>
> Thank you for your assessment and the many suggestions.
>
> ## Block MDP
> Thank you very much for this hint. The Block MDP formulation is indeed fitting here and we will incorporate it into the writeup accordingly.
>
> ## Combinatorial Hardness
> In this context, the term “combinatorial hardness” is used loosely. We refer to an environment as "combinatorially hard" if only very few of the exponentially many trajectories actually lead to the goal, while deviating from them often results in failure (e.g. *DigitJump* or *IceSlider*). Hence, some “intelligent” search algorithm is required. In this way, the process of retrieving a successful policy resembles that of a graph-traversing algorithm. While this is far from a formal definition, we will certainly mention this more clearly.
>
> ## Ablating $L_{FW}$
> Our world model relies on the combination of three loss-terms for training: $L_{FW}$, $L_{CE}$ and $L_{margin}$. We do not ablate $L_{FW}$ because setting its weight to zero would also set the gradient for all parameters of the forward model to zero, resulting in an untrained forward model, which would not be usable for planning.
>
> ## Ablating $L_{CE}$
> $L_{CE}$ is already ablated in Table 1 of the Appendix, where we show that planning performance decreases drastically without it.
>
> ## Ablating $L_{margin}$
> Note that the $\epsilon$ parameter of $L_{margin}$ also appears in the planner presented in Section 2.2. This double use of $\epsilon$ is intentional and critical. $L_{margin}$ can still be ablated, but this results in the need to estimate the latent margin $\epsilon$ for discretization during planning. To confirm this, we trained our model on *IceSlider* while setting the weight of $L_{margin}$ to zero. We then tried a grid of 4 values from 0.02 to 0.2 for $\epsilon$ and we observed that the planner is only effective for a narrow range of values, which is affected by the random seed. In general, removing $L_{margin}$ results in the lack of a margin that is consistent across the whole latent space and makes latent distances prone to more variance.
>
> ## Robust Success Metric
> We agree that such a metric could carry more meaningful information. As we would need to rerun all experiments to report this across all settings, we are currently only able to report preliminary results comparing PPGS and PPO when trained on 1000 levels in ProcgenMaze in the following table:
> ```
> |        |  Success Rate  |  Robust Success Rate  |
> |  PPGS  |     0.897      |         0.577         |
> |  PPO   |     0.667      |         0.496         |
> ```
>
> ## GS on Images
> We agree with your observation. GS on Images performs graph search in the observation space. It represents a good baseline as it can leverage the absence of visual noise and distractors. Since our environments do not stress this aspect, it performs reasonably well. On the other hand, this method is bound to fail as soon as we abandon such trivial cases: visual noise would render reidentification meaningless for GS on Images and the baseline would not be able to avoid revisiting vertices of the latent graph. We will add a discussion about visual noise etc. into the paper.
>
> ## More Training Levels
> Following your suggestion, we additionally benchmarked generalization across all environments for 2k, 5k, and 10k training levels. Results (success rates) are reported in the following table. We will update the plots in the paper.
> ```
> |       |     ProcgenMaze    |      IceSlider     |     DigitJump      |
> |       |  2k  |  5k  |  10k |  2k  |  5k  |  10k |  2k  |  5k  |  10k |
> |  PPGS | 0.850| 0.880| 0.880| 0.960| 0.965| 0.960| 0.960| 0.940| 0.980|
> |  PPO  | 0.746| 0.853| 0.880| 0.565| 0.590| 0.601| 0.513| 0.488| 0.492|
> ```
>
> ## Number of Training Samples
> PPGS is always trained on 400k samples for all experiments we report. As the number of training levels increases, fewer samples are collected in each level to keep the total number of samples constant. We will change our write-up to make this clear.
>
> ## Minor Comments
> Thank you for spotting and pointing out these issues. We will fix them accordingly. Regarding L280, we believe that the main disadvantage lies in only having access to few offline trajectories (while, for instance, PPO can collect 125x more samples on-policy). Not having access to a reward signal can also be seen as a drawback.

---

### Official Review · Reviewer_XKCd · 2021-07-31

**Rating:** 7
**Confidence:** 5

**Summary:**

In this work, the authors present a method to perform planning from pixels in environments that exhibit a hard combinatorial search space. They consider procedurally generated environments that can be modeled by a goal-conditioned MDP.  They present a method that first learns to represent the environment as a latent graph, in which nodes represent latent states and the edges represent actions, and then use a graph search algorithm to plan over this graph. They learn a state encoder model, parameterized by a ResNet, to project a pair of environment's states and actions to a d-dimensional hypersphere. They also learn a transition model over the latent space as well as an inverse model that is trained to prevent the loss of the forward model to collapse to a trivial solution. The authors also introduce an extra loss that enforces an epsilon margin between encodings of non-bisimilar states to ensure that the latent exhibits a discrete structure to enable planning. Then, they use the breadth-first search algorithm to find in the latent graph, for a given episode, the sequence of actions that lead from the initial state to the goal. They also use a Model Predictive Control approach to mitigate the prediction error propagation during planning. Finally, the authors introduce three visual environments that exhibit a combinatorial hard search space. They show that their method outperforms strong baselines on these environments and obtain good generalization.

**Limitations And Societal Impact:**

This paper doesn't add limitations or potential negative societal impact to existing reinforcement learning and classical planning methods.

**Main Review:**

First I would like to thank the authors for their work. I believe that learning to plan from pixels to solve combinatorial problems is very important and that this paper steps forward in that direction. My comment on the soundness of the claims, significance and novelty of the contribution, and relevance to the NeurIPS community are below:

The paper is very clear, well written and overall easy to follow. Notably,  I really enjoyed sections 1, 2 and 3 that are well motivated and didactic.

The method is sound and the design choices are clearly explained and motivated.

The experimental section is well-structured. The environments are clearly described.

The choice of the baselines makes sense and the supplementary material justifies this choice clearly.

The method seems to work really well and these environments compared to the baselines.

Limitations of this work:

In the related work, I would recommend the authors to additionally cite and compare their method to LEAP (Nasiriany et al.) that is really close to this method. LEAP would also be a good candidate for additional baselines.

While the way the model is trained and how the planning is performed are clearly described, I think this paper lacks a section to discuss how both are intertwined. Notably, if my understanding is correct, the model is first trained on randomly sampled transitions, then frozen and then the authors perform planning on this frozen model for all their experiments. This setting is interesting as it validates the representation capabilities of the model however it is limited. For instance, in some environments, some states and thus some transitions can't be visited by a random policy. Therefore, I think that it is important to also consider settings where the model is improved iteratively on transitions found during the search.

While these environments are complex along some axes, they remain simple along others. Notably, the dynamics seems simple enough to be learned from a few randomly sampled transitions. Therefore, I would recommend the authors test their method on other environments to strengthen their claims. For instance, the authors might consider the environments used in GLAMOR and assess if their method still outperforms GLAMOR on these.

The paper mentions several times the issue of generalization. While the authors addressed generalization on levels, it would be very interesting to address as well generalization on goals as the method allows for it. In this form, it is surprising when starting to read the experimental sections, to learn that the goals are not sampled but fixed. It would be great if the authors run additional experiments to assess generalization over goals. In this case, to ensure a FAIR comparison, they should also add the goal as an input to PPO's policy and DREAMER's model and use HER (Andrychowicz et al.) in addition.

To conclude, I think that this work is very promising and I really liked the paper. In this form, the paper is marginally above the acceptance threshold. However, I would be happy to increase my score if the authors address the limitations I mentioned.



**Time Spent Reviewing:**

5

---

> ### Author Response · Authors · 2021-08-10
> **Response to XKCd**
>
> We thank XKCd for the time invested in our paper and for all the constructive feedback.
> ## LEAP
> We agree that LEAP deals with a similar setup to ours and we will accordingly mention it in the related works. We found 3 main obstacles to implementing it as a baseline. Namely,
> 1. LEAP is designed for continuous control instead of discrete action spaces,
> 2. the goal-conditioned policy is trained on dense rewards requiring access to the true state of the environment and
> 3. representations are learned through a VAE on a fixed environment and simpler observations.
>
> Nonetheless, we adapted it to our setting by (1) replacing TD3 with Deep Q-Learning and (2) granting access to true states for computing rewards. We could not address issue (3), as a VAE is sensibly harder to train on a distribution of complex levels with respect to a PointMaze environment. Within the time span of this rebuttal we were limited to minimal tuning of parameters and architectures, mainly increasing the maximum number of steps in accordance with our evaluation procedure. We will perform an extended hyper-parameter optimization, however, due to (3) we do not expect performance close to PPGS. We report the preliminary success rates that we managed to achieve with our adapted implementation (1000 training levels).
> ```
> |        | ProcgenMaze | IceSlider | DigitJump |
> |  LEAP  |      -      |    0.08   |   0.20    |
> |  PPGS  |   0.897     |    0.97   |   0.99    |
> ```
>
> ## Iterative Model Improvement
> Thank you for this suggestion. We now conducted the following experiments: training our method starting from a small number of random trajectories and iteratively collecting on-policy data. We compared the performance of one-shot planning on *IceSlider* when trained on 1k levels. We measured success rates on unseen levels for (A) the default setting (400k random training samples), (B) in a low-data scenario (100k random training samples), and (C) when iteratively adding on-policy transitions to a small training set of 100k random training samples.
>
> In this last case, we collect 100k additional on-policy samples every 5 epochs. At 20 epochs we observe that on-policy data collection is able to accelerate and stabilize learning (see Table below). When training is completed (40 epochs), (A) and (C) reach the same performance, while (B) does not improve. We believe that this hints that
> 1. random trajectories are enough to eventually cover the state space of *IceSlider* well and
> 2. the planner can be effectively used for data collection and iterative improvements.
>
> We believe that collecting data on-policy in an RL-like loop is crucial in environments requiring more exploration and represents an interesting direction. We will add this to the final version of the paper.
>
> The success rates in *IceSlider*:
> ```
> epoch |(A)400k offline | (B)100k offline | (C)100k off + 300k on-policy |
>   20  |  0.331+-0.27   |   0.040+-0.01   |        0.778+-0.07           |
>   40  |  0.826+-0.05   |   0.043+-0.01   |        0.857+-0.04           |
> ```
>
> ## Other Environments
> GLAMOR is mainly benchmarked on continuous control and ATARI games, both of which do not involve a set of procedurally generated levels. Moreover, we believe that most ATARI environments do not represent combinatorially hard instances and are not a good fit for our method. While we intend to investigate our method's effectiveness on more domains, we are unable to do this in the context of this rebuttal.
>
> ## Generalization over Goals
> Test levels are completely unseen levels with unseen goal locations. The difficulty of achieving the environment goal varies largely across different levels. Sampling unseen levels at test time also implies achieving unseen goals in our opinion. Nevertheless, we additionally report success rates when sampling goals by random rollouts for each test level (see Table below). PPGS is goal-conditioned and reaching goals achieved by random policies is often easier than the unique goals we sample at test time.
> ```
> |      	     |   ProcgenMaze   |   IceSlider   |   DigitJump   |
> |   PPGS        |      0.85       |     0.94      |     0.99      |
> ```
> On the other hand, the baselines would require significant changes to allow goal-conditioning, which we cannot, unfortunately, fit in the scope of this rebuttal.
>
> Finally, we cannot refrain from celebrating this cool Pseudonym “xkcd”. The chances are one in 1/2 a million ;-)

---

> > ### Comment · Reviewer_XKCd · 2021-08-30
> > **Feedback following your response**
> >
> > Thank you for your thorough response! You have addressed my concerns. Given this additional information, I do think the paper is fit to be presented at the conference and I increase my score to 7. Regarding the pseudonym, yes it is a cool one indeed ;)

---

### Decision · Program_Chairs · 2021-09-27

**Decision:**

Accept (Poster)

**Comment:**

Learning dynamics models over pixels for model-based RL remains a largely open problem. There is consensus among the reviewers that the empirical results in this paper are impressive, and the approach constitutes a valuable contribution. The reviewers make several suggestions for improvements, e.g., analyzing generalization to goals and clarifying necessary conditions for environments it can perform well on, that can be addressed in a revision.